# Investigation of the Effect of Rice Bran Content on the Antioxidant Capacity and Related Molecular Conformations of Plant-Based Simulated Meat Based on Raman Spectroscopy

**DOI:** 10.3390/foods11213529

**Published:** 2022-11-06

**Authors:** Yanran Li, Ruisheng Jiang, Yuzhe Gao, Yumin Duan, Yifan Zhang, Minpeng Zhu, Zhigang Xiao

**Affiliations:** 1College of Grain Science and Technology, Shenyang Normal University, Shenyang 110034, China; 2Experimental Center of Shenyang Normal University (Department of Grain), Shenyang 110034, China

**Keywords:** rice bran, soybean protein isolate, Raman spectroscopy, meat substitute, high moisture extrusion, molecular conformations

## Abstract

At present, plant-based simulated meat is attracting more and more attention as a meat substitute. This study discusses the possibility of partial substitution of rice bran (RB) for soybean protein isolate (SPI) in preparing plant-based simulated meat. RB was added to SPI at 0%, 5%, 10%, 15%, and 20% to prepare RB-SPI plant-based simulated meat by the high moisture extrusion technique. RB-SPI plant-based simulated meat revealed greater polyphenol content and preferable antioxidant capacity (DPPH radical scavenging capacity, ABTS scavenging ability, and FRAP antioxidant capacity) compared to SPI plant-based simulated meat. The aromatic amino acids (tryptophan and tyrosine) of RB-SPI plant-based simulated meats tend to be masked first, and then the hydrophobic groups are exposed as RB content increases and the polarity of the surrounding environment increases due to the change in the disulfide conformation of RB-SPI plant-based simulated meats from a stable gauche–gauche–gauche conformation to a trans–gauche–trans conformation.

## 1. Introduction

The availability of high biological value proteins is becoming increasingly scarce due to the population’s rapid expansion. The trend of plant-based simulated meat replacing animal meat is quickly growing in response to consumer demand for a healthy diet, environmental pressures brought on by animal meat production, and concerns over animal welfare [1]. The industry for plant-based simulated meat is expanding quickly, and it is projected to produce $21 billion globally by 2025 [2], and this boom in new markets is anticipated to be sustained in the years to come.

Protein is essential for producing plant-based simulated meat, in addition to which additional functional substances need to be added to simulate the taste and texture of animal meat. Soybean protein isolate (SPI) is considered one of the high-quality raw materials for plant meat simulation because of its rich nutritional value, excellent performance, and low price [3]. Wheat gluten as a binder usually assists plant proteins in simulating the structure of meat fibers, thus improving the quality of the product to be as close as possible to the structure of animal meat [4]. To more accurately mimic the performance of meat products, creating fiber structures in plant-based simulated meat also requires specific processing technologies [5]. At present, high moisture extrusion is the most commonly used and widely studied technology that uses plant protein to form a meat-like fiber structure [6]. Raman spectroscopy is frequently used to study the spatial structure of proteins, which can be used to obtain information on changes in the backbone conformation of the protein molecule’s peptide chain and changes in the microenvironment of the molecular side chains [7]. By identifying changes in the characteristic peaks of the protein structure, one can predict changes in the structure of a protein molecule. Plant-based simulated meat is an excellent substitute for animal meat, which provides more sustainable and high-quality protein, but SPI has relatively lower methionine content and lacks the supply of dietary fiber that is beneficial to the human gut. Therefore, the need to find a substance that can partially replace SPI to make plants mimic meat to achieve a more diverse texture and better nutrition has attracted attention.

Rice bran (*RB*), a by-product of rice processing, has high nutritional value and is a rich source of protein, vitamins, polyphenols, minerals, and especially dietary fiber [8,9]. RB has great application value in the food industry, with a wide range of amino acids, and is rich in dietary fiber, which can provide unique structural and functional properties for processed food and enhance its nutritional value [10]. However, RB is typically employed to process animal feed [11], which unquestionably represents a significant loss of resources for high-quality agricultural and ancillary products. Most studies on plant-based simulated meat have focused on how different protein compounds affect the structure of the substance. Chiang Hong et al. (2018) [12] found that SPI plant-based simulated meat with 30% WG addition had better organizational and fibrous structure. RB, a common bran raw material, increased the content of soluble dietary fiber and its antioxidant capacity after extrusion treatment [13], so adding RB in plant-based simulated meat could complement SPI nutritionally. The dietary fiber and polyphenols contained in RB affect the side chain microenvironment in the molecular conformation of plant-based simulated meat proteins as well as the polypeptide backbone conformation, and understanding the changes in these molecular conformations has implications for subsequent studies on the fiber formation mechanism of high moisture extruded plant-based simulated meat. Zhang et al. [14] found that the increase in wheat gluten leads to the stretching of plant-based simulated meat polypeptide chains, which plays an essential role in promoting the formation of highly aligned fiber structures. However, there are few studies on the effect of RB on the performance of plant-based simulated meat, and the effect of RB content on the molecular conformation and function of plant-based simulated meat is uncertain.

Therefore, in this study, RB was co-extruded with SPI as an additive under high moisture extrusion to investigate the effect of RB content on the antioxidant capacity of plant-based simulated meat with SPI as the main raw material and to elucidate the changes in protein molecular conformation by Raman spectroscopy, which provides a new method for improving the nutritional value of plant-based simulated meat and a new strategy for the application of RB in the future, as well as a new way for further research on the fiber formation mechanism of plant-based simulated meat under extrusion.

## 2. Materials and Methods

### 2.1. Materials

SPI was provided by Kedong Yuwang Soybean Protein Food Co. (Qiqihar, China). RB was produced by Liaoning Shengbao Rice Co. (Shenyang, China). Grate the RB and pass it through an 80 mesh sieve before further use. The DPPH free radical scavenging ability kit and the ABTS free radical scavenging ability kit were purchased from Beijing Solaibao Technology Co. (Beijing, China). The gallic acid and total antioxidant ability (T-AOC) detection kit (FRAP microplate method) was purchased from Shanghai Yuanye Biotechnology Co. (Shanghai, China). Other analytical reagents were purchased from Tianjin Tianli Chemical Reagents Co. (Tianjin, China).

### 2.2. Extrusion Experiments

The extrusion experiment was performed using a lab-scale twin screw extruder (S56-Ⅲ, Jinan Saisin Extruding Machinery Co., Jinan, China). After extrusion stabilization, RB was screened through 80 mesh, and then raw materials were prepared with the mass ratios (*w*/*w*) of RB to SPI of 0%, 5%, 10%, 15%, and 20%. The extruder is divided into six regions, and the temperature of each area is set according to the experimental temperatures of Zhang et al. (2019) [15] and Fang et al. (2013) [16]. In this experiment, the temperature settings were the feeding zone (60 °C), mixing zone (80 °C), cutting zone (120 °C), cooking zone ⅰ (150 °C), cooking zone ⅱ (150 °C), and cooling zone (50 °C). In addition, in the extrusion experiment, the screw speed was set at 280 r/min, and the moisture content of the extrudate was controlled at 70% (*w*/*w*). Only the content of RB was changed, and other extrusion parameters were kept constant. After the extruded samples were cooled to room temperature, they were manually cut, put into sealed bags, and frozen at −18 °C. Samples were freeze-dried at −59 °C for 48 h, ground, and passed through a 100 mesh sieve.

### 2.3. Total Phenol Extraction and Content Determination

The extraction of total phenols from RB-SPI plant-based simulated meat was based on the method of Li’s research [17] and slightly modified. The lyophilized powder of RB-SPI plant-based simulated meat was accurately weighed at 0.3 g, added to 70% ethanol solution according to the material–liquid ratio of 1:30, and sonicated in a water bath at 40 °C for 1 h. An ultrasonic instrument (SB-25-12DTDN, Ningbo Xinzhi Biotechnology Co., Ningbo, China) was used in the extraction process. Centrifuge at 4000 r/min for 10 min, and collect the supernatant for later use.

The determination of total phenol content referred to the Folin–Ciocalteu method [18]. First, 1.0 mL of total phenol extract of RB-SPI plant-based simulated meat was taken in a 10 mL brown volumetric flask, 1 mL of forintol reagent was added, shaken well, and left for 1 min, then 3 mL of 7.0% Na_2_CO_3_ solution was added, shaken well and left for 1 min, and finally, the solution was fixed with distilled water, mixed well, and left for 90 min at 45 °C in a water bath protected from light, and measured by UV-1200S UV spectrophotometer at 760 nm. The standard curve was plotted with different gallic acid formulations (2, 4, 6, 8, 10, 12, 16 μg/mL), and the regression equation was Y = 0.0157x + 0.0507, R^2^ = 0.9985. The results were expressed in mg/g of gallic acid equivalent (GAE) per gram of sample.

### 2.4. Determination of Antioxidant Capacity

#### 2.4.1. DPPH Free Radical Scavenging Ability

The DPPH kit was used to determine the clearance according to the method described in the kit. Following the instructions of the kit, 25 μL of the sample solution was mixed and reacted thoroughly with 975 μL of the working solution. The mixed sample solution was kept in the dark for 30 min at room temperature, and the absorbance value was measured at 515 nm and recorded as A1. The rest of the mixture was treated as described above. The absorbance was measured using a mixture of 25 μL of sample solution and 975 μL of anhydrous ethanol as a control, denoted as A2. The absorbance was measured using a mixture of 25 μL of extract solution provided in the kit and 975 μL of working solution as a blank, denoted as A0, and the clearance of DPPH was calculated according to the following formula (1):DPPH free radical scavenging rate (%) = [1 − (A_1_ − A_2_)/A_0_] × 100(1)

#### 2.4.2. ABTS Free Radical Scavenging Ability

The ABTS kit was used for detection, and the clearance rate was obtained according to the kit instruction method. Following the instructions of the kit, 50 μL of the sample solution was reacted with 950 μL of the LABTS working solution. The rest of the mixture was treated as described above. After being fully mixed, the sample solution was left for 6 min at room temperature, dark, and the absorbance value was measured at 405 nm wavelength, denoted as A_1_. The absorbance was measured by equal volumes of reference solution replacing the working solution, denoted as A_2_; an equal volume of distilled water was substituted for the sample to measure the absorbance, denoted as A_0_, and the clearance rate of ABTS was calculated according to the following formula (2):ABTS free radical scavenging rate (%) = [1 − (A_1_ − A_2_)/A_0_] × 100(2)

#### 2.4.3. FRAP Antioxidant Capacity

The total antioxidant capacity (T-AOC) assay kit was used to determine the total antioxidant capacity according to the kit instruction method. The total antioxidant capacity was expressed by the concentration of Fe^2+^. The FRAP working solution was prepared according to the kit instructions, and the FRAP Assay buffer, TPTZ, and ferric chloride solution were configured in the ratio of 10:1:1, respectively. For the preparation of the Fe^2+^ standard gradient, the Fe^2+^ standard solution (10 mM) was diluted to 0.05, 0.1, 0.3, 0.5, 0.7, 0.9, and 1.2 mM with distilled water, and the solutions were added to the 96-well plate in sequence and mixed carefully. The 96-well plates were incubated in a shaking incubator at 37 °C for 30 min, and the absorbance values of the reaction solution at 630 nm were measured by an enzyme marker. The results were expressed as millimoles of Fe^2+^ per 100 g sample (mmol/100 g).

### 2.5. Ultraviolet–Visible (UV–Vis) Spectroscopic Analysis 

The 0.01 g RB-SPI plant-based simulated meat lyophilized powder was accurately weighed, 10 mL phosphate buffer (10 mmol/L, pH 7.0) was added, and stirred magnetically at 25 °C for 1 h, centrifuged at 8000 r/min for 10 min, and the supernatant was collected for detection. The absorption spectrum of RB-SPI plant-based simulated meat extract diluted by phosphate buffer solution was measured with the UV-9000 UV spectrophotometer (Shanghai Yuananalysis Instruments Co., Shanghai, China). The UV spectrophotometry conditions were slightly modified with reference to the method of research by Zhao et al. (2021) [19]. The scanning wavelength was 200–400 nm.

### 2.6. Fluorescence Spectroscopy Analysis

The fluorescence spectrum determination conditions refer to the method of research by Wu et al. (2020) [20]. A fluorescence spectrophotometer (F97, Shanghai Leng Light Technology Co., Shanghai, China) was used for fluorescence spectrum determination of RB-SPI plant-based simulated meat dilution in Step 2.5, with an excitation wavelength of 280 nm and a scanning wavelength of 280–460 nm. The scanning rate is 3000 nm/min, and the excitation slit and emission slit widths were set to 5 nm.

### 2.7. Raman Spectroscopic Analysis

RB-SPI plant-based simulated meat samples were determined by a Raman spectrometer (inVia, Renishaw, London, UK). The method was slightly modified from the research method by Lancelot et al. (2021) [21]. The spectral conditions are as follows: the excitation wavelength is set at 852 nm, the laser power is 300 mW, the scanning range is 400–2000 cm^−1^, and each scanning time is 60 s. The Raman spectra of the measured samples are plotted and output after signal accumulation and mean calculation. Raman spectrum processing: Labspec5 software was used for baseline correction and peak search. Phenylalanine (1003 cm^−1^) was used as the normalization factor to obtain the Raman spectra of plant-based simulated meat with different RB addition levels. PeakFit V4.12 was used to calculate the percentage of each configuration of the RB-SPI plant-based simulated meat disulfide bond.

### 2.8. Statistical Analysis

Data were obtained from experiments, and each group of data was repeated three times, expressed as mean ± standard deviation. Results Analysis SPSS Statistics 26 statistical software and Origin 2021 were used to analyze and plot the data. Significance was determined by one-way analysis of variance (ANOVA), followed by Duncan’s test. *p* < 0.05 was considered a statistically significant difference.

## 3. Results and Discussion

### 3.1. Effects of RB Content on Total Phenol Content and Antioxidant Capacity of RB-SPI Simulated Meat

As can be seen from Table 1, the plant-based simulated meat itself has certain antioxidant capacity. After extrusion, the soybean protein isolates gradually unfold, and the antioxidant amino acid residues originally embedded in the molecule are revealed to play an antioxidant role. With the increase of RB supplemental level, the total phenolic content, DPPH scavenging rate, ABTS scavenging rate, and FRAP antioxidant capacity of RB-SPI plant-based simulated meat increased gradually (*p* < 0.05). The polyphenol content and antioxidant capacity of modified gluten increased with the addition of RB, according to Wang et al. (2021) [22], which was compatible with the findings of our investigation. Compared with 0% RB, the total phenolic content, DPPH radical scavenging rate, ABTS radical scavenging rate, and FRAP antioxidant capacity of 20% RB, respectively, increased by 35.90%, 39.21%, 54.22%, and 85.71%. This may be because the RB-SPI plant-based simulated meat denatures under extrusion under high temperature, high pressure, and high shear, and phenolic compounds and polypeptide compounds containing phenolic groups in the complex increase. At the same time, the polyphenolic hydroxyl of RB binds to the soybean protein isolate to form stable groups, which promotes the improvement of antioxidant capacity of RB-SPI plant-based simulated meat [23].

### 3.2. Effects of RB Content on UV-Vis and Fluorescence Spectra of RB-SPI Simulated Meat

UV-vis spectroscopy is often used to study the influence of protein structure and intramolecular interaction. The UV-vis absorption spectrum of protein can initially investigate changes in protein molecular conformations because changes in the microenvironment of aromatic amino acid residues in protein molecules will change protein absorption wavelength [24]. The effect of RB on the tertiary structure of RB-SPI plant-based simulated meat proteins was analyzed using UV-vis spectroscopy. As can be seen from Figure 1a, the maximum absorption wavelength is 280 nm. With the increase of the RB supplemental level, the maximum absorption peak-to-peak value of the UV-vis absorption spectrum of RB-SPI plant-based simulated meat gradually increased. When the addition of RB was 20%, the absorption peak reached the maximum, and the wavelength of the maximum absorption peak shifted from 280 nm to 285 nm. This might be because, during extrusion, the hydrophobic groups in the protein molecules of the melt were exposed [25] and more combined with RB, making the microenvironment in which the aromatic amino acid residues were located decrease hydrophobicity and increase hydrophilicity. Therefore, the wavelength of the maximum absorption peak of the simulated meat in RB-SPI plants is redshifted.

Tryptophan (Trp), tyrosine (Tyr), and phenylalanine (Phe) are present in protein molecules. Proteins have endogenous fluorescence because of the presence of phenyl rings or conjugated double bonds that allow them to fluoresce at certain excitation wavelengths. Fluorescence spectrometry can detect the intrinsic fluorescence of proteins and is a sensitive method to study the conformation changes of proteins [26]. As shown in Figure 1b, at 280 nm excitation wavelength, RB-SPI plant-based simulated meat has the maximum fluorescence emission peak at about 350 nm. When the supplemental level of RB was 5%, the fluorescence intensity of RB-SPI plant-based simulated meat increased significantly, which might be because RB combined with the protein system of plant-based simulated meat to form a relatively stable complex, which made Trp in the state of inclusion, and the surrounding environment was mainly hydrophobic. With the increase of RB addition, the fluorescence intensity of RB-SPI plant-based simulated meat gradually decreased, which might be due to the fluorescence burst caused by the interaction of phenolic hydroxyl groups on polyphenols with tryptophan-based hydrophobic groups in the protein molecular conformation due to the unfolding of the plant-based simulated meat structure by rice bran addition. Our results are similar to those of Du et al. (2022) [27], who found that the addition of curcumin caused a significant fluorescence burst in SPI, which was attributed to the interaction of the phenolic ring in the polyphenol with the hydrophobic group. Typically, a blue shift in the maximum absorption wavelength indicates a decrease in the polarity of the surrounding environment and the burial of hydrophobic amino acids in the SPI, and conversely, a red shift indicates that the hydrophobic groups are exposed to a polar environment, and the protein molecules are unfolded [28]. The maximum absorption peak wavelength of RB-SPI plant simulants was red-shifted from 350 nm to 352 nm. This may be due to a large amount of RB interacting with the protein conformation of the plant-based simulated meat, increasing the polarity of the protein surroundings and exposing more Trp residues, resulting in a change in the maximum emission wavelength of the RB-SPI plant-based simulated meat [29].

### 3.3. Raman Spectroscopy Analysis of RB-SPI Plant-Based Simulated Meat

Raman spectroscopy is an effective tool for determining molecular conformational changes in protein systems, and microenvironmental changes in the conformation of protein molecules can be analyzed by changes in peak position and intensity in Raman spectroscopy. Quantitative information on changes in protein molecular structure can be expressed by the relative intensity of absorption peaks identified by protein molecular structure, that is, the ratio of absorption peak intensity identified by protein molecular structure to Phe absorption peak intensity [30]. The Raman spectra of the RB-SPI plant-based simulated meat at wavelengths of 400–2000 cm^−1^ are shown in Figure 2. The band at 1450 cm^−1^ can represent the bending vibration of aliphatic residues C-H, C-H_2_, and C-H_3_. The intensity of the 1450 cm^−1^ band in the RB-SPI plant-based simulated meat Raman spectrum increased first and then decreased with the increase of RB addition, indicating that the hydrophobic interaction of aliphatic residues increased [31]. In the wavelength range of 1630–1700 cm^−1^, the amide Ⅰ band of protein mainly came from C=O tensile vibration and out-of-plane C-N tensile vibration. Compared with the control, the stretching vibration of the group in the vegetated meat with RB was enhanced in this range, and the absorption was the strongest when the content of RB was 5%, and the absorption frequency moved to the lower wave number. This might be attributed to the increase in hydrogen bond strength [32]. The Raman spectra of plant base protein meat with different RB addition levels changed significantly, indicating that RB affects the spatial structure of plant-based protein meat and the environment of amino acid residues. The specific peak positions and vibration sources [33] are shown in Table 2.

#### 3.3.1. RB-SPI Simulated Meat Environmental Changes in Aromatic Amino Acids

Trp residues in proteins produce various Raman bands, among which the intensity of the Trp band located near 760 cm^−1^ is considered to be an indicator reflecting the hydrophobicity of indole rings [34]. Herrero (2008) [35] showed that the lower the intensity of the Raman band of Trp near 760 cm^−1^, the more the Trp of the protein tended to be exposed, otherwise, it tended to be embedded. As shown in Table 3, the intensity of the Trp spectrum of RB-SPI plant-based simulated meat increased firstly and then decreased with the increase of RB addition level. When the RB addition level was 5%, the Trp in the plant base meat tended to be embedded, and the Trp was buried in the hydrophobic microenvironment with reduced polarity. When the content of RB increased to 20%, the intensity of the I_760_ band decreased from 0.234 to 0.197, and Trp was exposed to the polar environment. The results are consistent with those of the fluorescence spectrum.

The 850 cm^−1^ and 830 cm^−1^ peaks of the protein Raman spectrum are related to the vibration of para-substituted benzene of Tyr residues, and the Tyr Fermi resonance line is the intensity ratio of the two peaks (I_850_/I_830_), which is commonly used to identify the degree of burial and exposure of Tyr components [36]. The phenolic hydroxyl group of the Tyr side chain can act as a donor or acceptor of hydrogen bonds. When the I_850_/I_830_ ratio is more than 1.0, Tyr residues are exposed to water and participate in the formation of moderate or weak hydrogen bonds. Conversely, a lower ratio indicates that Tyr residues are buried in a hydrophobic environment and tend to act as hydrogen bond donors to strengthen internal hydrogen bonds [37]. As shown in Table 3, the ratio of Tyr vibration peak band I_850_/I_830_ ranges from 1.244 to 1.396, indicating that the Tyr portion of the plant base meat with different RB addition levels is exposed to the polar microenvironment of the solution, and the OH group acts as the donor or acceptor of hydrogen bonds [38]. When the RB supplemental level increased from 0% to 20%, the ratio of I_850_/I_830_ increased, reflecting the gradual exposure of Tyr and the trend of more Tyr residues from hydrogen bond donors to acceptors [39].

#### 3.3.2. Changes in Disulfide Bridges Conformation of RB-SPI Plant-Based Simulated Meat

Disulfide bridges are the main force to maintain the tertiary structure in the protein, which can form both intramolecular disulfide bridges of protein and intermolecular disulfide bridges in the protein chain. The protein Raman spectrum of 500–550 cm^−1^ is the characteristic frequency of the stretching vibration of the disulfide bond, and the peak near 500–512 cm^−1^ can be classified as the gauche–gauche–gauche (g-g-g) configuration. The peak near 513–524 cm^−1^ is classified as gauche–gauche–trans (g-g-t) configuration, and the peak near 525–540 cm^−1^ is classified as trans–gauche–trans (t-g-t) configuration [40]. As can be seen from Table 4, the t-g-t mode is the main configuration of the disulfide bridge configuration of plant-based meat, followed by the g-g-g mode. When the RB addition level is 5%, the RB-SPI plant-based simulated meat disulfide bridges configuration is mainly transformed to the most stable energy g-g-g mode, and the content of intramolecular disulfide bridges increases while the intermolecular disulfide bridges decrease, indicating that RB and plant-based simulated meat form a stable structure [41]. With the increase of RB addition level, the disulfide bridge configuration of RB-SPI plant-based simulated meat changed from g-g-g mode to g-g-t and t-g-t mode. The network structure of the RB-SPI plant-based simulated meat was weakened when the RB addition level exceeded 5% [42], which may be attributed to the interaction between hydroxyl groups in RB fiber and water molecules through hydrogen bonds. RB-SPI plant-based simulated meat, by the limited contact between proteins and water, results in structural weakening [43].

## 4. Discussion

As shown in Appendix A, the figure includes the scoring and loading plots of the plant-based simulated meat. Principal component analysis (PCA) showed that 54.9% of the total variation was explained by the first principal component and 80.8% by the first two components, indicating a strong correlation between the antioxidant capacity, molecular conformation, and RB addition of RB-SPI plant-based simulated meat. The proximity to each other in the same quadrant indicates a high positive correlation. In the same direction, each parameter showed independent variation. The images showed that the scoring plot showed that the 0% RB was in a different direction from the other groups with RB addition, indicating that there was a significant difference between the RB-added group and the non-RB-added group. The fact that the 5% RB and 10% RB groups were in the same quadrant suggests that there was not much difference between them. The photos revealed that the most significant variables for the first component (54.9%), which exhibited a positive connection, were RB addition, total phenolic content, DPPH free radical scavenging, ABTS free radical scavenging, and FRAP antioxidant capacity. Disulfide bond content, I_760_, and I_860_/I_830_ made up the majority of the second component (25.9%).

The first component showed that the antioxidant capacity of RB-SPI plant-based simulated meat was highly correlated with the RB content, especially the DPPH radical scavenging rate showed a strong correlation with the rice bran content, which was due to the polyphenols contained in the RB providing electrons to pair with the single electrons in DPPH, thus scavenging the DPPH radical. As can be seen from the scoring plot, the five well-differentiated clusters were dispersed from left to right with increasing RB addition, indicating that the addition of RB increased the total phenolic content and antioxidant capacity of RB-SPI plant-based simulated meat and reached the highest value at 20% RB addition. The antioxidant capacity of RB-SPI plant-based simulated meat was positively correlated with the unstable t-g-t in the disulfide bond structure and negatively correlated with I_760_ was negatively correlated, because excess RB caused the soybean isolated to denature and unfold under extrusion conditions, forming more unstable t-g-t structures. The unfolding of the protein was accompanied by the disruption of the hydrophobic core, exposing hydrophobic amino acids (e.g., tryptophan and tyrosine) that could react with oxidants [44], increasing the antioxidant capacity of the RB-SPI plant-based simulated meat. The second component showed that the g-g-g and g-g-t structures in the disulfide bond content of RB-SPI plant-based mock meat were negatively correlated with the t-g-t structure. Additionally, there was a negative correlation between I_760_ and I_850_/I_830_. Combined with the scoring plot, it is clear that RB has a strong positive correlation with the more stable g-g-g and g-g-t structures at 5% RB addition, which is because 5% RB addition causes a change in the molecular conformation of RB-SPI plant-based simulated meat, resulting in a stable complex when the tryptophan in the protein molecule is encapsulated, resulting in a higher ratio of I_760_.

Strong correlations were found between the quantity of RB added and the antioxidant capacity and molecular conformation of the RB-SPI plant-based simulated meat. The polyphenols in rice bran were the main factor affecting the antioxidant capacity of RB-SPI plant-based simulated meat. The changes in its protein molecular conformation were mainly the result of the influence of the g-g-g and t-g-t structures in the disulfide bond. RB-SPI plant-based simulated meat had high antioxidant capacity and a more stable conformation, while the changes in the molecular conformation of the plant-based simulated meat induced by the amount of rice bran added were interesting. This study suggests that RB shows the potential to partially substitute SPI in plant-based simulated meat production.

## 5. Conclusions

This study investigated the effect of RB content on the antioxidant capacity of RB-SPI plant-based simulated meat and the change of protein molecular conformation using Raman spectroscopy. The antioxidant capacity and total phenolic content of the plant-based meat were both boosted by the addition of RB, which is rich in polyphenols, and the total phenolic content of the plant-based simulated meat with a 20% RB addition increased by 35.90%. The aromatic amino acids in RB-SPI plant-simulated meat tended to be embedded initially and then exposed when the RB content was increased. RB first reacted with SPI to form a stable complex that embedded the aromatic amino acids into the hydrophobic environment. The further reaction of RB with SPI at the addition of more than 5% increased the polarity of the surrounding environment exposing Trp and Tyr to the microenvironment, which was associated with the change of the vibrational mode of the disulfide bond from a stable g-g-g conformation to a less stable t-g-t conformation. The creation of plant-based simulated meat products with RB additions, as well as the optimization of product formulations, will benefit from the research on the impact of RB additions on the antioxidant capacity and molecular conformation of RB-SPI plant-based simulated meat.

## Figures and Tables

**Figure 1 foods-11-03529-f001:**
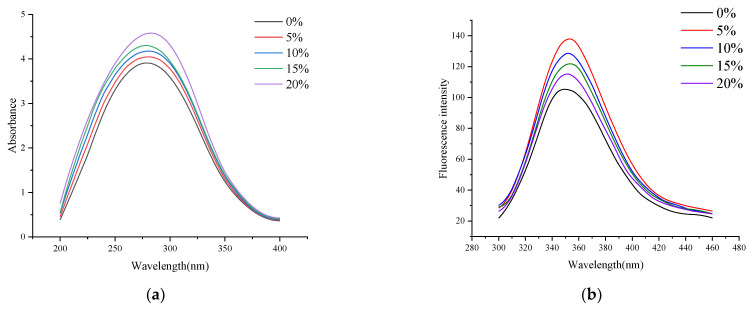
UV-vis spectra (**a**) and fluorescence spectra (**b**) of RB-SPI plant-based simulated meat with different RB contents.

**Figure 2 foods-11-03529-f002:**
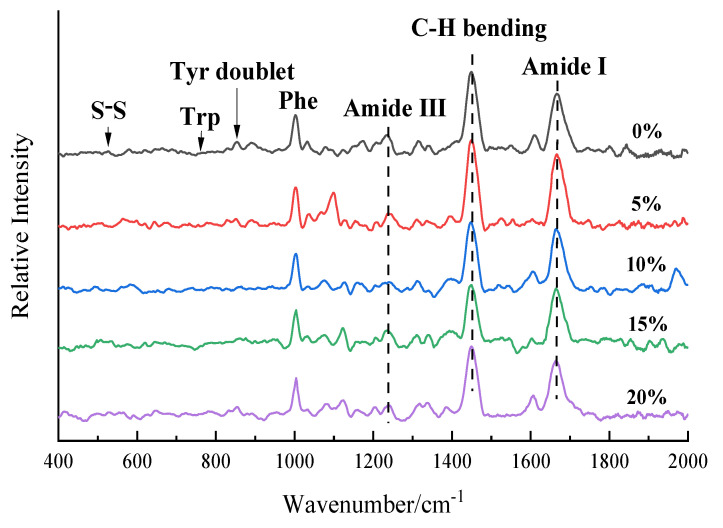
RB-SPI plants simulated meat Raman spectra with different RB contents.

**Table 1 foods-11-03529-t001:** The effect of RB content on the total phenolic content and antioxidant capacity of RB-SPI plant-based simulated meat.

RB Addition (%)	Total Phenolic Content (mg GAE/g)	DPPH Radical Scavenging Method (%)	ABTS Radical Scavenging Method (%)	FRAP Antioxidant Capacity (mmoL/100 g)
0	10.89 ± 0.30 ^d^	8.11 ± 0.08 ^c^	38.18 ± 0.74 ^e^	1.40 ± 0.07 ^d^
5	11.94 ± 0.39 ^c^	8.66 ± 0.18 ^c^	44.07 ± 1.03 ^d^	1.87 ± 0.02 ^c^
10	13.22 ± 0.01 ^b^	9.58 ± 0.37 ^b^	53.17 ± 0.53 ^c^	2.04 ± 0.06 ^bc^
15	14.13 ± 0.05 ^ab^	10.27 ± 0.45 ^b^	56.31 ± 0.87 ^b^	2.27 ± 0.02 ^b^
20	14.80 ± 0.64 ^a^	11.29 ± 0.18 ^a^	58.88 ± 1.03 ^a^	2.60 ± 0.02 ^a^

Note: Data are expressed as means ± standard of triplicate measurements. GAE, gallic acid equivalent; DPPH, diphenyl picryl hydrazinyl radical; ABTS, 2,2′-Azinobis-(3-ethylbenzthiazoline-6-sulphonate); FRAP, ferric ion reducing antioxidant power. Mean values in the same column with different lowercase are significant difference (*p* < 0.05).

**Table 2 foods-11-03529-t002:** Assignment of bands in the Raman spectrum of RB-SPI plant-based simulated meat.

Wave Number/cm^−1^	Peak Assignment
500–550	S-S stretching vibration
760	Trp residues
830	Tyr residues
850	Tyr residues
1003 ± 1	Phe residues
1220–1330	Amide III band
1420–1480	Aliphatic side chain C-H bend
1630–1700	Amide I band

Note: Trp, tryptophan; Tyr, tyrosine; Phe, phenylalanine.

**Table 3 foods-11-03529-t003:** Band intensities of protein side chain groups in RB-SPI plant-based simulated meat.

Amount of RB Added (%)	Trp Band Intensity I_760_	Tyr Vibration Peak Doublet Ratio I_850_/I_830_
0	0.201 ± 0.112 ^a^	1.396 ± 0.197 ^a^
5	0.234 ± 0.087 ^a^	1.244 ± 0.210 ^a^
10	0.207 ± 0.050 ^a^	1.255 ± 0.086 ^a^
15	0.201 ± 0.048 ^a^	1.259 ± 0.138 ^a^
20	0.197 ± 0.043 ^a^	1.264 ± 0.114 ^a^

Note: Different letters in the same column indicate significant differences between groups, *p* < 0.05. The normalized intensity is the ratio of the relative intensity of each Raman band to the intensity of the phenylalanine band at 1003 cm^−1^.

**Table 4 foods-11-03529-t004:** Changes of disulfide bridges configuration in RB-SPI plant-based simulated meat.

Amount of RB Added (%)	Disulfide Bridges Structure Content (%)		
	g-g-g	g-g-t	t-g-t
0	31.86 ± 0.78 ^b^	29.28 ± 4.09 ^ab^	38.87 ± 4.87 ^ab^
5	43.64 ± 4.41 ^a^	31.65 ± 4.24 ^ab^	24.71 ± 0.17 ^c^
10	38.80 ± 3.29 ^ab^	34.77 ± 1.41 ^a^	26.43 ± 1.89 ^c^
15	35.68 ± 5.34 ^ab^	30.03 ± 4.04 ^ab^	34.29 ± 1.29 ^bc^
20	30.30 ± 2.81 ^b^	24.62 ± 3.40 ^b^	45.09 ± 6.22 ^a^

Note: g-g-g, gauche–gauche–gauche; g-g-t, gauche–gauche–trans; t-g-t, trans–gauche–trans. Different letters in the same column indicate significant differences between groups, *p* < 0.05.

## Data Availability

Data is contained within the article or Appendix A.

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
