# Peer review of "Investigation of the Effect of Rice Bran Content on the Antioxidant Capacity and Related Molecular Conformations of Plant-Based Simulated Meat Based on Raman Spectroscopy"

_foods, 2022, doi:10.3390/foods11213529_

Round 1
Reviewer 1 Report
The topic addressed is very interesting and actual but the way of the presentation should be improved.
Please adapt the reference format in the text, according to journal Authors' guidelines.
Lines 58 - 66 - Authors specified there are few studies related to the ``effects of RB on the properties of plant simulated meat, and the effects of RB on the structure and function of plant simulated meat are still unknown.`` - here you cite only 2 papers. Cite more in order to argue better this paragraph or rephrase it.
Line 76 - please rephrase: Grate the RB and pass through an 80 mesh sieve before further use. (now it seems like working instructions, not a method description).
The English language needs style improvement. Several spelling and grammar errors were found. Please check it carefully.
Lines 74-82 - please carefully write the sentences by using the proper punctuation.
Line 96 - no data about the freeze-drying procedure was given. The study could not be replicated in this way of description. Carefully give all the details necessary.
Line 116 - what do you mean with DPPH kit? Give more details about this method.
Line 126 - what do you mean with ABTS kit? Give more details about this method.
Line 136 - total antioxidant capacity (T-AOC) assay kit ? Give more details about this method.
Line 149 - please give all details related to equipment producer and country of production.
Please use the same Font in the entire manuscript.
There are abbreviations in tables that are not described within the footnotes. Please use abbreviations only after you first mentioned their descriptions in the text.
Please pay attention there are many phrases without verbs (e.g. Exposing Trp and Tyr to microenvironment. - line 332).
Author Response
Reply to Reviewer #1
Reviewer #1:
The topic addressed is very interesting and actual but the way of the presentation should be improved.
ANS: Thank you for your suggestion! We had carefully improved the presentation as requested by the reviewers.
Please adapt the reference format in the text, according to journal Authors' guidelines.
ANS: Thank you for your suggestion! We had revised the format of the references in accordance with the journal's requirements.
Lines 58 - 66 - Authors specified there are few studies related to the ``effects of RB on the properties of plant simulated meat, and the effects of RB on the structure and function of plant simulated meat are still unknown.`` - here you cite only 2 papers. Cite more in order to argue better this paragraph or rephrase it.
ANS: Thank you for your suggestion! We had added relevant literature to this section for additional clarification. Plant simulated meat are usually prepared by mixing various proteins in a certain ratio (Joshi, & Kumar, 2015; Kumar et al., 2017), and most studies on plant simulated meat have explored the effect of protein ratios on their structural properties, such as Jiang et al. who found that SPI plant simulated meat supplemented with 30% WG had better organization and fiber structure. Whereas, bran is a common ingredient that promotes the formation of fibrous structure in meat (Ke & Hsieh, 2007; Delcour et al., 2012). Therefore, it would be an interesting study to investigate the effect of dietary fiber-rich bran-based ingredients (e.g., rice bran) on the properties of plant-based vegetarian meat.
Joshi, V. K., & Kumar, S. (2015). Meat Analogues: Plant based alternatives to meat products-A review. International Journal of Food and Fermentation Technology, 5(2), 107-119. https://doi.org/10.5958/2277-9396.2016.00001.5
Kumar, P., Chatli, M. K., Mehta, N., Singh, P., Malav, O. P., & Verma, A. K. (2017). Meat analogues: health promising sustainable meat substitutes. C R C Critical Reviews in Food Technology, 57(5), 923-932. https://doi.org/10.1080/10408398.2014.939739
Ke, S. L., & Hsieh, F. H. (2007). Protein-protein interactions in high moisture-extruded meat analogs and heat-induced soy protein gels. Journal of the American Oil Chemists Society, 84(8), 741-748. https://doi.org/10.1007/s11746-007-1095-8
Delcour, J. A., Joye, I. J., Pareyt, B., Wilderjans, E., Brijs, K., & Lagrain, B. (2012). Wheat gluten functionality as a quality determinant in cereal-based food products. Annual Review of Food Science & Technology, 3(1), 469-492. https://doi.org/10.1146/annurev-food-022811-101303
Line 76 - please rephrase: Grate the RB and pass through an 80 mesh sieve before further use. (now it seems like working instructions, not a method description).
ANS: Thank you for your suggestion! This sentence had been changed to "Samples were freeze-dried at -59°C for 48h, ground, and passed through a 100 mesh sieve"
The English language needs style improvement. Several spelling and grammar errors were found. Please check it carefully.
ANS: Thank you for your suggestion! We had checked and corrected spelling and grammatical errors.
Lines 74-82 - please carefully write the sentences by using the proper punctuation.
ANS: Thank you for your suggestion! We had carefully revised the punctuation.
Line 96 - no data about the freeze-drying procedure was given. The study could not be replicated in this way of description. Carefully give all the details necessary.
ANS: Thank you for your suggestion! We had supplemented the data on the temperature and time of freezing-drying.
Line 116 - what do you mean with DPPH kit? Give more details about this method.
ANS: Thank you for your suggestion! The DPPH kit is a quick and easy-to-use test for the scavenging ability of the DPPH radical. The principle of the test is that the DPPH radical is a relatively stable nitrogenous radical, and its solution is purple when dissolved in ethanol, with strong absorption at 517nm. When the radical scavenger is present, the degree of discoloration of the solution is proportional to the number of electrons accepted, so the absorbance decrease can be used to reflect the nitrogen radical scavenging ability of the sample. We had added the operation of this method to the manuscript.
Line 126 - what do you mean with ABTS kit? Give more details about this method.
ANS: Thank you for your suggestion! The ABTS kit is a method that uses ABTS as a chromogenic agent to detect the antioxidant capacity of a solution. The principle of this method is that ABTS is oxidized to green ABTS+ in the presence of an oxidant. The production of ABTS+ is inhibited in the presence of an antioxidant, and the antioxidant capacity of the sample can be calculated by measuring the absorbance of the solution at 405 nm. We had added some parameters for the use of this method.
Line 136 - total antioxidant capacity (T-AOC) assay kit ? Give more details about this method.
ANS: Thank you for your suggestion! The total antioxidant capacity assay kit is a method for determining the antioxidant activity of samples using Fe3+ reduction capacity. The method is based on the principle that the reduction of Fe3+-TPTZ by antioxidant substances in the sample under acidic conditions produces Fe2+-TPTZ, which exhibits a distinct blue-purple color with maximum light absorption at 593 nm. The reduction capacity of the sample, i.e. the total antioxidant capacity, can be obtained by measuring the amount of blue substance produced. We had added the operational details of this method in the paper.
Line 149 - please give all details related to equipment producer and country of production.
ANS: Thank you for your suggestion! We had added to the manuscript information about the manufacturer and production of the device.
Please use the same Font in the entire manuscript.
ANS: Thank you for your suggestion! The same font had been used for the article.
There are abbreviations in tables that are not described within the footnotes. Please use abbreviations only after you first mentioned their descriptions in the text.
ANS: Thank you for your suggestion! We had referred to the abbreviations in the table in the text.
Please pay attention there are many phrases without verbs (e.g. Exposing Trp and Tyr to microenvironment. - line 332)
ANS: Thank you for your suggestion! We had carefully corrected these errors.
Reviewer 2 Report
It would have been interesting to have done HPLC studies to quantify the main phenolic compounds present in the sample, not only as total phenolics.
Author Response
Reply to Reviewer #2
Reviewer #2:
It would have been interesting to have done HPLC studies to quantify the main phenolic compounds present in the sample, not only as total phenolics.
ANS: Thank you for your suggestion! The quantification of the main phenolic compounds present in the samples using HPLC methods is indeed a very interesting study. We have made some changes to the title and preface of the article, which focuses on the effect of rice bran content on the antioxidant activity of plant simulated meat and the analysis of its structural changes by Raman spectroscopy, so this method may not quite fit the focus of this article. We will take your suggestions fully into account in future studies and thank you again for your valuable suggestions!
Reviewer 3 Report
Title: “Nutritional functions” ? or functional properties? Or nutraceutical characterization?
Comment: Article not include extrusion parameters nor any discussion on extrusion, so I think that author must erase “produced by high moisture extrusion cooking technology extrusion” from title.
Include proximal analysis for RB-SPI samples.
Figure 1: figure 1 title not correspond to graphs a) and b).
Line 230-225. Author must include references to probe its discussion. It is very hard discuss some effect in a dispersion curve as figure 1 b).
Structural characterization must include, DRX, SEM, FTIR y DSC. This article did not have this kind of characterization.
If author wants to do research in “Nutritional functionality” then author must include digestibility and/or biodisponiblity for protein.
Article is focused on applying RB-SPI Raman Spectroscopy to RB-SPI. So, I think that author must focus this article toward something like Conformational Stability of soy bean protein addded with rice bran by Raman Spectroscopy. Or something like that.
Phenols analysis are not “nutritional functions”. Functional properties are reported in: https://doi.org/10.1016/j.lwt.2016.09.012.
Author Response
Reply to Reviewer #3
Reviewer #3:
Title: “Nutritional functions” ? or functional properties? Or nutraceutical characterization?
ANS: Thank you for your suggestion! The initial title of the manuscript was "Nutritional functions" because RB is rich in nutrients (Yu et al., 2012) which can increase the total phenolic content and antioxidant activity of RB-SPI plant simulated meat. After discussion, we had changed the title from "nutritional function" to "antioxidant activity".
Yu, C. W., Luo T., Xie T., Li, J., & Deng, Z. Y.(2022). Classified processing of different rice bran fractions according to their component distributions.International Journal of Food and Fermentation Technology, 57(7),4052-4064. https://doi.org/10.1111/ijfs.15715
Comment: Article not include extrusion parameters nor any discussion on extrusion, so I think that author must erase “produced by high moisture extrusion cooking technology extrusion” from title.
ANS: Thank you for your suggestion! The experimental RB-SPI plant simulated meat preparation was prepared using a high-moisture extrusion method. In this experiment, the temperature Settings are the feeding zone (60 ℃), mixing zone (80 ℃), cutting zone (120 ℃), cooking zone ⅰ (150 ℃), cooking zone ⅱ (150 ℃), cooling zone (50 ℃). The screw speed was set at 280 r/min, and the moisture content of the extrudate was controlled at 70% (w/w), Only the content of RB was changed, and other extrusion parameters were kept constant. The setting of extrusion parameters in this paper was supported by prior experiments. Taking temperature as an example, our initial extrusion temperature experiments were conducted in the range of 130~170°C with a 10°C gradient, using hardness as a reference basis. the hardness of RB-SPI plant simulated meat showed a trend of increasing and then decreasing with increasing temperature, reaching a maximum at 150°C. When the temperature was 170°C, the RB-SPI plant mock meat could not be stably extruded, which is the same situation as that of the high moisture extruded soy protein reported by Liu et al. The innovation of this paper is the use of rice bran and soybean isolate protein for high moisture extrusion, which expands the application channels of rice bran and increases the nutritional source of plant simulated meat, and further explores the relationship between rice bran addition, antioxidant activity and protein structure. Further basic optimization studies are considered as the next step of this paper and are not consistent with the focus of this paper, so they are not reflected in this paper. We hope to have your understanding and support!
Liu, K. S., & Hsieh, F. H. (2008). Protein-protein interactions during high-moisture extrusion for fibrous meat analogs and comparison of protein solubility methods using different solvent systems. Journal of Agricultural and Food Chemistry, 56(8), 2681–2687. https://doi.org/10.1021/jf073343q
Include proximal analysis for RB-SPI samples.
ANS: Thank you for your suggestion! The focus of this article is on exploring the relationship between rice bran addition, antioxidant activity, and protein structure, the proximal analysis may not quite fit the topic of the article, but our subsequent work will take your suggestions into full consideration, thanks again for your suggestion!
Figure 1: figure 1 title not correspond to graphs a) and b).
ANS: Thank you for your suggestion! We had carefully revised it.
Line 230-225. Author must include references to probe its discussion. It is very hard discuss some effect in a dispersion curve as figure 1 b).
ANS: Thank you for your suggestion! We had supplemented the literature in the manuscript counterpart to support our experimental results. Regarding the fluorescence spectra, RB addition of more than 5% causes a decrease in the fluorescence intensity of RB-SPI plant simulated meat, and the experimental results are attributed to a fluorescence burst caused by the interaction of phenolic hydroxyl groups on rice bran polyphenols with tryptophan-dominated hydrophobic groups, similar to the experimental results of Du et al. (2022). The maximum absorption peak wavelength of RB-SPI plant-based mock meat produces a red shift, which indicates that the hydrophobic group is exposed to a polar environment and the protein molecule is unfolded (Li, & Wang, 2015).
Du, C., Xu, J., Luo, S., Li, X., Mu, D., Jiang, S., & Zheng, Zhi. (2022). Low-oil-phase emulsion gel with antioxidant properties prepared by soybean protein isolate and curcumin composite nanoparticles. LWT-Food Science and Technology,161,113346.
Li, J., & Wang, X. (2015). Binding of (−)-epigallocatechin-3-gallate with thermally-induced bovine serum albumin/ι-carrageenan particles. Food Chemistry, 168, 566–571.
Structural characterization must include, DRX, SEM, FTIR y DSC. This article did not have this kind of characterization.
ANS: Thank you for your suggestion! We had focused the article on the structure of RB-SPI plant simulated meat based on Raman analysis, so these structural characterization methods are not consistent with the focus of this paper and therefore do not appear in the article, but will be considered for the next future studies. Thanks again for your suggestions!
If author wants to do research in “Nutritional functionality” then author must include digestibility and/or biodisponiblity for protein.
ANS: Thank you for your suggestion! We had changed the title from "nutritional function" to "antioxidant activity" and will consider the protein digestibility of RB-SPI plant simulated meat in subsequent studies. Thanks again for your suggestions!
Article is focused on applying RB-SPI Raman Spectroscopy to RB-SPI. So, I think that author must focus this article toward something like Conformational Stability of soy bean protein addded with rice bran by Raman Spectroscopy. Or something like that.
ANS: Thank you for your suggestion! We had focused the article on the structural analysis of RB-SPI plant mock meat based on Raman spectroscopy, with changes in the title, abstract, preface, and conclusions. We had changed the title to "Antioxidant activity and structural properties based on Raman spectroscopy analysis of rice bran-soybean isolate protein plant simulated meat produced by high moisture extrusion cooking technology". Among the Raman spectroscopy analyses are the exposure of hydrophobic groups such as tryptophan and tyrosine, secondary structure analysis, and the content of each disulfide bond. We have supplemented the article with a secondary structure analysis of the RB-SPI plant simulated meat, as detailed in article 3.3.1. Thank you again for your suggestion!
Phenols analysis are not “nutritional functions”. Functional properties are reported in: https://doi.org/10.1016/j.lwt.2016.09.012
ANS: Thank you for your suggestion! We had used the total phenolic content as a supplement to the antioxidant activity content rather than as a nutritional function.
Round 2
Reviewer 3 Report
Title is about structure , so author must include structural characterization. Structural characterization must include, DRX, SEM, FTIR y DSC. This R1 article do not have this kind of characterization.
Title include extrusion so author must develop and study the effect of the extrusion on RB-SPI. Author must include extrusion experimental design.
There is not any conclusion about extrusion.
Author did not do the suggestions to improve article, Author leave it to future.
In this R1 version, author mention a lot of future activities, really author must do it now to improve article.
Discussion about peptides and disulfide bridges are very good, but it is not enough and it is not the scope of this journal.
Author Response
Reply to Reviewer #3
Reviewer #3:
Title is about structure , so author must include structural characterization. Structural characterization must include, DRX, SEM, FTIR y DSC. This R1 article do not have this kind of characterization.
ANS: Thank you for your suggestion! We have carefully thought through the title to make it fit with the core research content of the article, and the revised title is "Investigation of the effect of rice bran content on the antioxidant activity and related molecular conformations of plant simulated meat based on Raman spectroscopy". In addition to this, we have added a discussion of molecular conformation in the text, and these changes serve the new title as well as the content of the article.
The structural characterization does include the detection of magnetic resonance imaging, scanning electron microscopy, and Fourier transform infrared spectroscopy, but the focus of our article is to investigate the effect of RB content on the molecular conformation of RB-SPI plant simulated meat by UV spectroscopy, fluorescence spectroscopy, and Raman spectroscopy, and to analyze the feedback of protein tertiary structure changes on primary structure, and the analysis of these results can reflect the effect of RB addition on protein moieties, microenvironmental hydrophobicity and disulfide bond conformational changes. The effect of fluorescent dyes on the structure of wet gluten was studied only by Nawrocka et al. [1] using fluorescence spectroscopy and Raman spectroscopy, and they combined these two spectra to analyze the interaction between fluorescein and gluten proteins. Therefore, considering the title of the article as well as the content of the study, these experiments for structural characterization are out of our scope for this article. Previous articles by our team have described experiments on structural characterization related to RB-SPI plant simulated meat [2].
- Nawrocka, A., Rumi´nska, W., Szyma´nska-Chargot, M., Niewiadomski, Z.,& Mi´s,A. (2022). Effect of fluorescence dyes on wet gluten structure studied with fluorescence and FT-Raman spectroscopies. Food Hydrocolloids, 131,107820.
https://doi.org/10.1016/j.foodhyd.2022.107820
- Jiang, , Xiao,Z., Huo, J., Wang, H., Li, H., Su, S., Duan, Y., & Gao, Y. (2022). Effects of rice bran content on plant-based simulated meat: From the aspects of apparent properties and structural characteristics. Food Chemistry,380,131842.
https://doi.org/10.1016/j.foodchem.2021.131842
Title include extrusion so author must develop and study the effect of the extrusion on RB-SPI. Author must include extrusion experimental design. There is not any conclusion about extrusion.
ANS: Thank you for your suggestion! We have revised the title based on the content of the article, and the changed title is "Investigation of the effect of rice bran content on the antioxidant activity and related molecular conformations of plant simulated meat based on Raman spectroscopy". The innovation of this paper is to explore the effect of RB content on the antioxidant activity and protein molecular conformation of RB-SPI plant simulated meat. The extrusion temperature was set as feeding zone (60°C), mixing zone (80°C), cutting zone (120°C), cooking zone i (150°C), cooking zone ii (150°C), and cooling zone (50°C). The screw speed is set to 280r/min and the moisture content of the extrudate is controlled at 70% (w/w), only the content of RB is changed and other extrusion parameters remain unchanged. Therefore exploring the effect of extrusion parameters on the properties of RB-SPI plant simulated meat is not in the scope of this paper, and not much discussion was done about extrusion, and this part is not consistent with the focus of the article. The article focuses more on the effect of rice bran content on plant simulated meat. Further basic optimization studies are considered as the next step in this paper. In order to further investigate the formation of molecular conformation of RB-SPI plant simulated meat during extrusion, we will fix the parameters of rice bran addition and change the parameters of extrusion temperature, water addition and screw speed in the future. We hope to get your understanding and support!
Author did not do the suggestions to improve article, Author leave it to future. In this R1 version, author mention a lot of future activities, really author must do it now to improve article.
ANS: Thank you for your suggestion! We were initially inspired by the articles of Gul et al [1] and Wang et al [2] to understand that rice bran is rich in polyphenols and presents high antioxidant activity, however, few studies have applied RB to plant simulated meat, so the focus of our article is to investigate the effect of rice bran addition on the antioxidant activity of RB-SPI plant simulated meat and the use of Raman spectroscopy to probe the conformation of protein molecules changes in the molecular conformation of the protein. We further analyze the relationship between RB-SPI plant simulated meat and antioxidant activity in the discussion of the article in 4. We focus on the antioxidant activity of RB-SPI plant simulated meat, and the experimental aspects such as protein digestibility will be the next step of the article and are not considered in this paper. We hope we can get your understanding and support!
1.Gul, K., Yousuf, B., Singh, A. K., Singh, P., & Wani, A. A. (2015). Rice bran: Nutritional values and its emerging potential for development of functional food - A review. Bioactive Carbohydrates and Dietary Fibre, 6(1), 24-30.
http://dx.doi.org/10.1016/j.bcdf.2015.06.002.
2.Wang, Z., Ma, Y., Chen, H., Deng, Y., Wei, Z.,Zhang, Y., Tang, X., Li, P., Zhao, Z., Zhou, P., Liu, G., & Zhang, M. (2022). Rice bran-modified wheat gluten nanoparticles effectively stabilized pickering emulsion: An interfacial antioxidant inhibiting lipid oxidation. Food Chemistry, 387,132874.
https://doi.org/10.1016/j.foodchem.2022.132874
Discussion about peptides and disulfide bridges are very good, but it is not enough and it is not the scope of this journal.
ANS: Thank you for your suggestion! We added a principal component analysis of the RB-SPI plant simulated meat at "4. Discussion". We further explored the relationship between antioxidant activity, molecular conformation and rice bran addition in RB-SPI plant simulated meat, and the PCA showed strong correlation between these three variables. Among them, the rice bran addition of more than 5% caused the soybean isolate to denature and unfold under extrusion conditions, forming more unstable t-g-t structures. The unfolding of the protein was accompanied by the disruption of the hydrophobic core, exposing hydrophobic amino acids (e.g. tryptophan and tyrosine) that could react with oxidants, which in turn increased the antioxidant activity of RB-SPI plant simulated meat.